# FastLongSpeech: Enhancing Large Speech-Language Models for Efficient Long-Speech Processing

**Shoutao Guo**[1,3], **Shaolei Zhang**[1,3], **Qingkai Fang**[1,3], **Zhengrui Ma**[1,3],
**Min Zhang**[4], **Yang Feng**[1,2,3*]

[1]Key Laboratory of Intelligent Information Processing,
Institute of Computing Technology, Chinese Academy of Sciences (ICT/CAS)
[2] Key Laboratory of AI Safety, Chinese Academy of Sciences
[3] University of Chinese Academy of Sciences, Beijing, China
[4] School of Future Science and Engineering, Soochow University
guoshoutao22z@ict.ac.cn,zhangshaolei20z@ict.ac.cn,fengyang@ict.ac.cn

## Abstract

The rapid advancement of Large Language Models (LLMs) has spurred significant progress in Large Speech-Language Models (LSLMs), enhancing their capabilities in both speech understanding and generation. While existing LSLMs often concentrate on augmenting speech generation or tackling a diverse array of short-speech tasks, the efficient processing of long-form speech remains a critical yet underexplored challenge. This gap is primarily attributed to the scarcity of long-speech training datasets and the high computational costs associated with long sequences. To address these limitations, we introduce FastLongSpeech, a novel framework designed to extend LSLM capabilities for efficient long-speech processing without necessitating dedicated long-speech training data. FastLongSpeech incorporates an iterative fusion strategy that can compress excessively long-speech sequences into manageable lengths. To adapt LSLMs for long-speech inputs, it introduces a dynamic compression training approach, which exposes the model to short-speech sequences at varying compression ratios, thereby transferring the capabilities of LSLMs to long-speech tasks. To assess the long-speech capabilities of LSLMs, we develop a long-speech understanding benchmark called LongSpeech-Eval. Experiments show that our method exhibits strong performance in both long-speech and short-speech tasks, while greatly improving inference efficiency [2].

## 1 Introduction

Benefiting from the advancement of Large Language Models (LLMs) [1–3], Large Speech-Language Models (LSLMs) have also made significant strides by extending the speech capabilities of LLMs. By harnessing the knowledge and reasoning abilities of LLMs, LSLMs can directly comprehend speech signals, perform analysis and reasoning, and achieve superior performance in a diverse of tasks such as speech recognition, speech translation, and speech understanding [4–6]. The ability to process and understand diverse speech signals has emerged as a key research focus in LSLMs [7].

To handle speech inputs, traditional methods [8, 9] typically employ a cascaded pipeline, where speech is first transcribed into text and then processed by LLMs. However, these approaches suffer from error propagation and discard valuable paralinguistic information [10]. To overcome these drawbacks, recent research [11–13] has shifted towards an end-to-end paradigm, enabling LSLMs

---

[*]Corresponding author: Yang Feng.
[2]The Code is at `https://github.com/ictnlp/FastLongSpeech.git`.

39th Conference on Neural Information Processing Systems (NeurIPS 2025).

to directly process and reason with speech signals. These methods can be broadly divided into two categories. On one hand, some approaches [4, 14–16] such as Qwen2-Audio [5] align the output spaces of pre-trained audio encoders with the embedding of LLMs, which allows speech inputs to be accommodated while transferring partial capabilities of the employed LLMs. At the same time, other methods [11, 17] involves discretizing the speech to discrete units, which allows LSLMs to handle speech units similar to text tokens. Despite these advancements, current LSLMs are largely constrained to processing short speech segments, typically under 30 seconds [5, 18]. Only a few LSLMs [12] have achieved a processing duration of 30 minutes on speech summarization tasks by relying on the construction of extensive, specialized training datasets.

The processing of long-form speech by LSLMs remains a largely unexplored area, primarily due to two significant challenges. First, unlike the abundance of diverse short-speech datasets [19–21], there is a scarcity of training data specifically for long-speech alignment and instruction, and the generation of such data is costly. Second, long-speech sequences impose substantial computational demands on LSLMs. The sequence of speech representations is often more than four times longer than its text equivalents for the same content [17]. which, in the context of long-form speech, leads to significantly higher computational costs. Therefore, LSLMs face considerable challenges in modeling long-speech sequences, stemming from both a scarcity of training data and increased computational costs. These challenges limit the exploration and application of LSLMs in long-speech processing.

To address the above challenges, we propose FastLongSpeech, a novel framework designed to extend the capabilities of LSLMs to long-speech processing, leveraging only short-speech training data. We utilize Qwen2-Audio[3] [5], the currently representative LSLM, as our foundational speech-language model. To enable long-speech processing, FastLongSpeech incorporates a speech extractor module on top of the audio encoder [22]. This module employs our proposed iterative fusion strategy to compress the speech representations output by the audio encoder, preserving essential temporal information while reducing redundancy. The resulting condensed speech representations are then used by LLM for comprehension and reasoning. In our method, the speech extractor significantly reduces the sequence length of the condensed representations, thereby lowering the computational costs for subsequent LLM.

To further adapt LSLM for long-speech processing, FastLongSpeech employs a two-stage training approach. In the first stage, a CTC loss [23] is introduced within the extractor module to measure the text density of the input speech representations, which is utilized for the iterative fusion strategy. The second stage introduces a dynamic compression training method, enabling LLM to effectively adapt to condensed speech representations. This stage leverages existing short-speech data and dynamically adjusts the compression ratios in the iterative fusion strategy to transfer the understanding and reasoning capabilities of LSLMs to long-speech tasks.

To evaluate the long-speech understanding capabilities of LSLMs, we also develop a benchmark, called LongSpeech-Eval. Experiments show that FastLongSpeech achieves efficient speech processing on both long-speech and short-speech benchmarks, and can balance efficiency and effectiveness to meet different requirements.

## 2 Background

Our method builds upon Qwen2-Audio [5] and incorporates CTC loss [23]. We provide a brief overview of these key components below.

**Qwen2-Audio** Qwen2-Audio is a large-scale audio-language model capable of comprehending various audio signals, performing reasoning, and generating text responses. It consists of three modules: an audio encoder [22], an LLM [24], and an adaptor. The adaptor serves to align the output of the audio encoder with the embeddings of LLM, enabling the LLM to process speech inputs. Following extensive pre-training, supervised fine-tuning, and DPO [25], Qwen2-Audio demonstrates remarkable performance across a wide range of speech tasks. Given a raw waveform $\mathbf{s} = (s_1, ..., s_N)$ sampled at 16 kHz, the audio encoder produces a sequence of speech representations $\mathbf{h} = (h_1, ..., h_J)$

---

[3]For convenience, Qwen2-Audio refers to Qwen2-Audio-7B-Instruct throughout this paper.

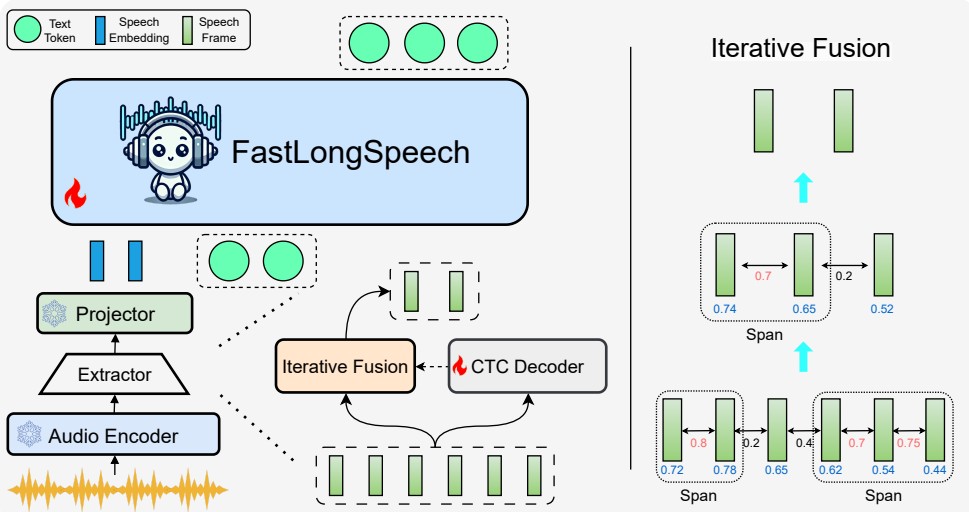

Figure 1: Architecture of FastLongSpeech. The left panel illustrates that FastLongSpeech generates a response based on the input speech and text instruction. The right panel details the iterative fusion strategy, where numbers between adjacent frames denote similarity scores and numbers below frames represent content density.

with 25 Hz frame rate. The LLMs then generate the text response $\mathbf{y}$ based on $\mathbf{h}$ and an instruction $\mathbf{x}$:

$$p(\mathbf{y}|\mathbf{x}, \mathbf{h}) = \sum_{i=1}^{I} p(y_i \mid \mathbf{y}_{<i}, \mathbf{x}, \mathbf{h}), \tag{1}$$

where $p(y_i \mid \mathbf{y}_{<i}, \mathbf{x}, \mathbf{h})$ denotes probability distribution of the next token.

**CTC** Connectionist Temporal Classification (CTC) [23] is widely used in Automatic Speech Recognition (ASR), where the input is typically longer than the corresponding output [26]. To align speech with the transcript, CTC introduces a blank token into the vocabulary and defines the possible output as an alignment $\mathbf{a}$. Each time step in this alignment corresponds to either a blank token or a non-blank token. The alignment has the same length as the speech sequence and can be reduced to the final transcript through a collapse function $\Gamma^{-1}$. During training, CTC employs an efficient dynamic programming algorithm to maximize the probability of all possible alignments corresponding to the ground-truth transcript $\mathbf{c}$:

$$\mathcal{L}_{ctc} = -\log \sum_{\mathbf{a} \in \Gamma(\mathbf{c})} p_{ctc}(\mathbf{a} \mid \mathbf{h}), \tag{2}$$

where $\Gamma(\mathbf{c})$ denotes the set of all alignments corresponding to $\mathbf{c}$, and $\mathbf{h}$ denotes the sequence of speech representations produced by audio encoder [22]. In this paper, we leverage the output distribution of CTC to quantify the text density of speech representations $\mathbf{h}$, which is subsequently utilized in the iterative fusion strategy.

## 3 Method

In this section, we introduce the architecture of FastLongSpeech, with a particular emphasis on its novel speech extractor module. To empower LSLM with the ability to learn speech compression techniques and adapt to long-speech processing, we present an innovative two-stage training approach. Additionally, we introduce the LongSpeech-Eval benchmark, designed to evaluate the long-speech understanding capabilities of LSLMs.

## 3.1 Model Architecture

Figure 1 illustrates the model framework of FastLongSpeech. Building upon Qwen2-Audio, we incorporate an advanced extractor module, which features a CTC [23] decoder and employs our proposed iterative fusion strategy to condense speech representations. The workflow of FastLongSpeech proceeds as follows. Given a waveform $\mathbf{s}$ sampled at 16 kHz, the audio encoder converts it into a mel-spectrogram and processes it through convolution and transformer layers [22], yielding a sequence of speech representations $\mathbf{h}$ with 25 Hz frame rate. Subsequently, the extractor module processes $\mathbf{h}$ using an iteration fusion strategy to produce the condensed representations $\mathbf{h}'$, whose length is within the speech window of LLM. The speech window denotes the maximum length of the speech representations $\mathbf{h}$ during training, specifically the maximum number of speech frames. LLM then utilizes the condensed representations $\mathbf{h}'$ along with the instruction $\mathbf{x}$ to generate the response $\mathbf{y}$, as described in Eq.(1).

## 3.2 Iterative Fusion

To facilitate efficient long-speech processing, we introduce an extractor to compress lengthy and sparse speech representations [17] into more compact forms. Our primary objective is to minimize information loss during compression, preserving essential temporal information while reducing redundancy. Thus, our method needs to retain speech representations that contain more textual content, while discarding excessively similar adjacent speech frames. To achieve this, we propose an iterative fusion strategy that incrementally merges selected representations based on content density and similarity between adjacent frames, ultimately yielding condensed representations.

We first define the metrics to measure content density and frame similarity, followed by an introduction of our iterative fusion strategy. For a given speech frame $h_j$, content density is generally associated with the amount of textual information it contains [27, 28]. To quantify this, we employ a CTC decoder, whose output distribution provides the probabilities of the speech frame being classified as either a blank or a non-blank token. Consequently, the content density of $h_j$ is derived from the sum of probabilities for non-blank tokens:

$$d_j = \sum_{a_j \neq \epsilon} p_{ctc}(a_j \mid h_j), \tag{3}$$

where $\epsilon$ denotes the blank token. Frame similarity, on the other hand, captures the overlap in content between adjacent speech frames. We measure this using cosine similarity between $h_j$ and $h_{j+1}$:

$$e_{j,j+1} = \frac{h_j h_{j+1}}{|h_j||h_{j+1}|}. \tag{4}$$

After introducing these measurements, we provide an overview of our iterative fusion strategy in Figure 1. For $m$-th iteration, we first determine the length $T(m)$ of current speech representations and the length for the next iteration:

$$T(m+1) = \begin{cases} \lfloor T(m)/2 \rfloor, & \text{if } T(m) > 2L \\ L, & \text{if } T(m) \leq 2L \end{cases}, \tag{5}$$

where $L$ denotes the final target length of condensed representations $\mathbf{h}'$. The number of speech frames to be reduced is then calculated as:

$$r(m) = T(m) - T(m+1). \tag{6}$$

Subsequently, we utilize the similarity metric between adjacent frames, as defined in Eq.(4) to identify the $r(m)$ most similar pairs of adjacent frames. The consecutive identified frames are grouped into a span, as illustrated in Figure 1. For each span, we employ a weighted fusion approach, leveraging the content density in Eq.(3) as weights to merge all frames within the span into a single compressed speech frame. This process yields a new sequence of speech representations with length $T(m+1)$, comprising both the newly condensed frames and the remaining uncompressed frames. If $T(m+1)$ still exceeds the target length $L$, we initiate another iteration. Otherwise, the resulting speech representations from this round constitute the final condensed representations $\mathbf{h}'$.

This iterative fusion strategy effectively reduces the length of speech representations by half in each iteration, facilitating efficient speech processing.

### 3.3 Training Method

After introducing the iterative fusion strategy, we further present a two-stage training method to enable FastLongSpeech to perform long-speech tasks. To facilitate the generation of condensed representations through the iterative fusion strategy, the first stage of training focuses on the ASR task, allowing FastLongSpeech to learn the measurement of content density. Building on this, the second stage incorporates our proposed dynamic compression training method, which helps LLMs adapt to short-speech condensed representations with varying compression ratios, thereby transferring short-speech capabilities to long-speech processing.

**CTC Training**  In the first stage, we aim to leverage the ASR task to enable FastLongSpeech to recognize the amount of textual information in speech representations, namely content density. To achieve this, we introduce a CTC decoder in the extractor module, which is trained using the CTC loss [23] as shown in Eq.(2). This allows us to utilize the generation distribution of the CTC decoder to measure the content density of speech representations. In this stage, we only train the CTC decoder.

**Dynamic Compression Training**  In the second stage, we introduce a novel dynamic compression training method. The introduction of this method is based on two considerations. First, it enables the LLM to adapt to condensed representations $\mathbf{h}'$ with different compression ratios. Second, the current <s, x, y> triplet training data primarily contains short-speech clips, which are typically under 30 seconds in duration [29]. By sampling the length $L$ of condensed representations [30], LSLM can maintain its perception of speech sequences corresponding to the length of its speech window, without avoiding excessive bias towards overly condensed speech sequences. The dynamic compression training method is as follows:

$$L_{dct} = -\sum_{L \sim \mathcal{U}(\mathrm{L})} \log p(\mathbf{y} \mid \mathbf{x}, \mathrm{IF}(\mathbf{h}, L)), \tag{7}$$

where $L$ is uniformly sampled from L, which is a set of hyperparameters. $\mathrm{IF}(\mathbf{h}, L)$ represents applying the iterative fusion operation on the speech representations $\mathbf{h}$ to obtain the condensed representations of length $L$. After the training process, FastLongSpeech can transfer the short-speech capabilities of LSLMs to long-speech tasks.

### 3.4 LongSpeech-Eval Benchmark

After introducing FastLongSpeech, we aim to evaluate its capacity to handle long-speech inputs. Due to the lack of benchmarks for assessing the long-speech capabilities of LSLMs, we construct LongSpeech-Eval. This benchmark is based on the MultiFieldQA-En and NarrativeQA subsets of LongBench [31], a comprehensive long context understanding benchmark. This benchmark assesses the ability of LLM to answer questions based on the long document.

To construct LongSpeech-Eval, we aim to convert the document into speech. We first employ Llama3.1-70B-Instruct[4] to filter out samples containing numerous formulas or non-English characters. Subsequently, GPT-4o [32] is utilized to summarize and polish the document into a spoken format. Llama3.1-70B-Instruct is then used to answer questions based on the spoken-form document, with inappropriate samples manually discarded. For the remaining samples, we then synthesize speech for the spoken-form document using Orca[5]. Consequently, each sample in the LongSpeech-Eval consists of the synthesized speech along with the corresponding questions and answers. More details are provided in **Appendix A**.

## 4 Experiments

### 4.1 Datasets

For the training data in the first stage, we utilize the ASR data, which contain 960 hours of LibriSpeech [33] data and 3k hours of data sampled from MLS [34]. In the second training stage, our training data primarily originates from three datasets following the Spoken QA format: **OpenASQA** [35],

---

[4]https://huggingface.co/meta-llama/Llama-3.1-70B-Instruct
[5]https://github.com/Picovoice/orca.git

**LibriSQA** [36], and **Common Voice** [37]. All employed samples used for training are under 30 seconds in duration. For OpenASQA, we use the Open-Ended Speech AQA subset (5.9k hours), which covers diverse question types including content, speaker style, and emotion. For LibriSQA, we incorporate the 360-hour training set. For Common Voice, we adapt the English subset (1.7k hours) to the spoken QA format by generating transcription instructions via ChatGPT and using the original transcripts as the ground-truth answers.

For evaluation, we employ a diverse set of nine datasets spanning five distinct tasks to comprehensively assess performance:

**Short-Speech Spoken QA**: We utilize three datasets: the speech_QA_iemocap (AIR-Bench) [38], the LibriSQA test set [36], and the LibriTTS test subset from OpenASQA [35]. The three datasets contain rich set of QA pairs involving paralinguistic information. We utilize this task to evaluate the effectiveness of various speech fusion methods under different compression ratios.

**Long-Speech Spoken QA**: We leverage our proposed LongSpeech-Eval benchmark to assess the performance of different methods in long-speech understanding scenarios.

**Spoken Dialogue Understanding**: We evaluate the inference efficiency of our method using speech_dialogue_QA_fisher subset from AIR-Bench.

**Emotion Recognition**: We leverage the MELD dataset [39] to benchmark our method against other efficiency method [40] under diverse efficiency scenarios.

**ASR**: We use the LibriSpeech [33] test-clean and test-other, and GigaSpeech [41] test set to evaluate ASR performance. The ASR task utilizes short-speech samples, which evaluate the capacity to extract complete content from the speech.

**Speech Information Retrieval**: We introduce SPIRAL-H benchmark, which is introduced in the paper of SpeechPrune [42] for long speech information retrieval.

More details of the training and evaluation dataset are in **Appendix B**.

## 4.2 System Settings

Given the current scarcity of long-speech LSLMs, we implement several baseline approaches in addition to our FastLongSpeech.

**Random** method randomly selects frames from speech representations and arranges them sequentially to serve as conditioning input for LSLM.

**AvgPool** method sequentially selects a fixed number of speech frames to form segments and then apply average operation within each segment.

**MostSim** method first selects the most similar consecutive speech frames and then applies the average pooling operation to each set of adjacent representations.

**NTK-RoPE** method modifies the base Rotary Position Embedding (RoPE) [43] of LLMs to an NTK-Aware Scaled RoPE [44], thereby extending the speech window of Qwen2-Audio to match the context length of its LLM. This method is exclusively utilized for long-speech reasoning tasks.

**Cascaded** first uses Whisper-Large-V3 [22] to transcribe the audio, and then pass the resulting context text to Qwen-7B-Chat (since Qwen2-Audio-7B is based on Qwen-7B-Chat).

**Baseline** refers to the direct application of vanilla Qwen2-Audio for inference, which is not employed in long-speech inference tasks.

**FastLongSpeech** refers to our proposed method.

We use Qwen2-Audio-7B-Instruct[6] as the base LSLM for all methods, with the length of speech window as 750. Besides, we also extend our method to vanilla Qwen2.5-Omni [15] without dynamic compression training to verify the effectiveness of our method. In the first stage of training for our FastLongSpeech, we only train the CTC decoder. In the second stage, we experimentally set $\mathbf{L}$ to {750, 400, 200, 100, 50, 25, 12} and fine-tune the LLM of Qwen2-Audio using LoRA [45]. For a fair comparison, we also fine-tune Qwen2-Audio for all methods except the Baseline and FastLongSpeech,

---

[6]`https://huggingface.co/Qwen/Qwen2-Audio-7B-Instruct`

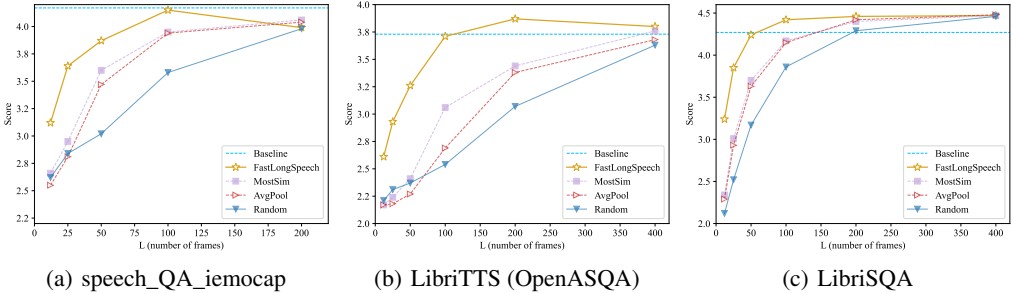

| (a) speech_QA_iemocap | (b) LibriTTS (OpenASQA) | (c) LibriSQA |

Figure 2: Performance of diverse speech fusion methods in the short-speech spoken QA tasks. The score is derived from the LLM evaluating the quality of responses based on the questions and ground-truth answers. The baseline model utilizes a speech window of 750 frames. For the methods other than Baseline, we regulate the compression ratio by adjusting the target length $L$ of the condensed speech representations. In the experiments, a smaller value of $L$ corresponds to a higher compression ratio. A higher score indicates a better quality of the responses.

using the same training data and implementation settings as used for FastLongSpeech. All methods employ the original prompt template from Qwen2-Audio. For more training details, please refer to the **Appendix C**.

During evaluation, we use greedy search for all methods and control compression ratios by adjusting the target length $L$. For long-speech inference, we split the input speech into a series of 30-second clips, which are processed by the audio encoder and then combined into a complete sequence of speech representations in temporal order. To evaluate the performance, we employ various metrics tailored to each task. For the Spoken QA and Spoken Dialogue Understanding task, we use Llama3.1-70B-Instruct [2] to score responses on a scale of 1 to 5, with the scoring template available in the **Appendix D**. For the ASR task, we use Word Error Rate (WER) to assess the accuracy of the generated transcripts. For Emotion Recognition task, we use the Accuracy (ACC) metric to evaluate the performance.

### 4.3 Main Results

We evaluate the performance of our method on short-speech and long-speech spoken QA tasks.

For the short-speech spoken QA task, Figure 2 illustrates the performance of various methods across three datasets. Our Fast-LongSpeech method consistently outperforms other methods on all three datasets under different speech compression ratios, maintaining high response quality even at a 30-fold compression ratio ($L = 25$). Unlike the Random method, which arbitrarily discards speech representations, other methods consider all temporal information when compressing speech representation, resulting in improved generation quality [29]. Compared to AvgPool and MostSim methods, our method more effectively eliminates redundant information while preserving highly informative speech representations, leading to better performance across various com-

Table 1: Performance of various speech methods on long-speech spoken QA task.

| Method | Score ($\uparrow$) |
|---|---|
| Random | 2.54 |
| Similar | 3.08 |
| AvgPool | 3.10 |
| NTK-RoPE | 3.44 |
| **FastLongSpeech** | **3.55** |

pression ratios. Notably, when $L$ equals 12, other speech fusion methods exhibit similarly suboptimal performance, while our method maintains a substantial performance advantage. We attribute this superiority to our novel iterative fusion strategy and dynamic compression training approach. Furthermore, compared to vanilla Qwen2-Audio, our method achieves comparable performance with a shorter sequence of speech representations, demonstrating higher efficiency.

For the long-speech spoken QA task, our method outperforms other approaches in generation quality, as evidenced in Table 1. To handle the long-speech input, methods such as Random, Similar, and AvgPool employ their respective fusion techniques to compress the speech representations within the speech window. However, these approaches yield suboptimal generation quality, primarily due to ineffective fusion strategies and misaligned training methods. In contrast, NTK-RoPE expands the

speech window of LSLM to the context length of LLM, thereby preserving more speech information and achieving improved performance. Furthermore, our method leverages a more effective speech fusion strategy coupled with a dynamic compression training approach, transferring the short-speech reasoning capabilities of LSLMs to the long-speech domain. Notably, despite utilizing the same speech window size as Qwen2-Audio [5], our method achieves optimal performance in long-speech comprehension tasks with greater efficiency than NTK-RoPE.

## 5 Analysis

To provide a comprehensive evaluation of our approach, we conduct extensive analyses. We then introduce each analytical experiment in detail.

### 5.1 Ablation Study

To gain a comprehensive understanding of the contributions made by different components in our approach, we conduct detailed ablation experiments. As shown in Table 2, both the iterative fusion and dynamic compression training strategies proposed in FastLongSpeech significantly enhance the performance of LSLMs on long-speech reasoning tasks. First, the dynamic compression training strategy effectively transfers the short-speech capabilities of LSLMs to long-speech scenarios, utilizing only short-speech data. This approach enables LLMs to adapt to condensed representations at varying compression ratios and mitigate over-reliance on excessively compressed speech representations. Consequently, Fast-LongSpeech can compress long-speech representations to fit within the speech window length, facilitating efficient long-speech processing at high compression ratios. Moreover, multiple iterations in the iterative fusion approach lead to substantial improvements in generation

Table 2: The ablation experiments of our method on long-speech benchmark. "w/o DCT" replaces Dynamic Compression Training method with standard fine-tuning approach. "w/o Iterative Fusion" eliminates the multiple iterations in the iterative fusion. "w/o Content Density" substitutes the method of merging all speech frames within the same span with an average pooling operation.

| Method | Score (↑) |
|---|---|
| **FastLongSpeech** | **3.55** |
| w/o DCT | 3.33 |
| w/o Iterate Fusion | 3.41 |
| w/o Content Density | 3.28 |

quality. This finding underscores the benefits of progressively expanding the receptive field [46] in iterative fusion for aggregating semantic information. Furthermore, guided by content density, our iterative fusion strategy tends to retain more informative speech frames [27], resulting in the most significant performance improvement.

### 5.2 Inference Efficiency

After investigating the impact of various components in FastLongSpeech, we conduct analyses on the inference efficiency across different methods. To quantify this efficiency, we employ the TFLOPs metric, which measures the average number of floating-point operations (FLOPs) across the entire dataset and is calculated using calflops[7] tool. For long-speech scenarios, we incorporate average runtime as an additional efficiency indicator, which is measured in seconds. Table 3 and 4 present the results of inference efficiency experiments, which are obtained on NVIDIA L40.

Table 3: The inference efficiency on LibriTTS test subset of OpenASQA dataset, where "Ours" denotes FastLongSpeech.

| Method | Score (↑) | TFLOPs (↓) |
|---|---|---|
| Baseline | 3.73 | 9.79 |
| Ours ($L$=400) | 3.80 | 8.54 |
| Ours ($L$=200) | 3.87 | 5.64 |
| Ours ($L$=100) | 3.71 | 4.17 |

In short-speech tasks, our approach demonstrates performance comparable to vanilla Qwen2-Audio while requiring only half the computational resources. When the allocated computational resources are increased, we can achieve better results. Notably, the computational costs decrease as the compression ratio increases. This not only demonstrates the better efficiency of our model but also highlights its ability to balance generation quality and inference efficiency.

---

[7]https://github.com/MrYxJ/calculate-flops.pytorch

The advantages of our method become even more pronounced in long-speech tasks, where our method achieves better generation quality than NTK-ROPE, with a 70% reduction in runtime and a 60% decrease in computational costs. Compared to the cascaded method, it even achieves a speedup of more than sevenfold, underscoring its sub-

Table 4: The efficiency on the long-speech benchmark.

| Method | Score ($\uparrow$) | TFLOPs ($\downarrow$) | Time ($\downarrow$) |
|--------|-------|--------|------|
| NTK-RoPE | 3.44 | 61.21 | 4.80 |
| Cascaded | **3.75** | n/a | 17.23+1.38 |
| **Ours** | 3.55 | **26.44** | **1.47** |

stantial efficiency advantage for processing long-form speech. This further shows the effectiveness of our method in handling long-speech inputs. For spoken dialogue understanding, emotion recognition and speech information retrieval tasks, please refer to the Appendix E and F.

## 5.3 Content of Condensed Representations

Beyond the spoken QA, spoken dialogue understanding and emotion recognition tasks, we extend our evaluation to the ASR task, which requires precise transcription of the entire speech content [47]. Through this task, we explore variations in condensed representations across different compression ratios. Table 5 demonstrates the ASR performance of Qwen2-Audio and our method. At low compression ratios ($L$=400), FastLongSpeech performs comparably to Qwen2-Audio, demonstrating the effectiveness of our dynamic compression training and iterative fusion strategy in preserving speech content. Unlike Qwen2-Audio, our method does not require substantial post-processing to extract the transcript, with strong instruction following abilities. At higher compression ratios ($L$=100), FastLongSpeech slightly trails Qwen2-Audio in ASR but maintains comparable results

Table 5: The performance on the ASR task, where "Ours" denotes the Fast-LongSpeech. For the dataset, "Clean" and "Other" denote LibriSpeech test-clean and test-other sets. "Giga" denotes the test set of GigaSpeech. The results are evaluated with WER metric.

| Method | Clean | Other | Giga |
|--------|-------|-------|------|
| Baseline | 3.85 | 6.70 | 13.71 |
| Ours ($L$=750) | 4.04 | 7.02 | 11.76 |
| Ours ($L$=400) | 4.08 | 7.17 | 11.77 |
| Ours ($L$=200) | 4.36 | 7.40 | 12.70 |
| Ours ($L$=100) | 27.12 | 24.61 | 23.69 |

in spoken QA, as illustrated in Figure 2. This indicates that while our approach demonstrates applicability across diverse tasks, the optimal compression ratio is inherently task-dependent. Therefore, achieving an effective balance between efficiency and effectiveness thus necessitates careful calibration and a thorough assessment of resource constraints.

## 6 Related Work

**Large Speech-Language Models** With the advancements in Large Language Models (LLMs), recent research attempts to extend the understanding and reasoning capabilities of LLMs to speech inputs, becoming Large Speech-Language Models (LSLMs). Early studies [8, 9] employ a cascading paradigm, where speech is first transcribed into text before being processed by LLMs. More recently, some works [5, 48, 14, 12] utilize the adaptors to align the output space of speech encoders with the input space of LLMs, achieving multi-task LSLMs. Other approaches [49, 11, 50, 17] utilize speech discretization techniques, converting waveforms into discrete units, enabling LSLMs to process speech in the same way they process text. These approaches allow LSLMs to handle both speech understanding and generation.

**Long Sequence Modeling** Long sequence modeling presents challenges across diverse domains, including text, video, and speech. The approaches to long-context modeling vary depending on the type of the inputs. For extended text sequences, researchers explored methods such as position interpolation and extrapolation [51], sliding window [52], continuous fine-tuning on long-text data [53], and native sparse attention [54]. To address long-video processing, recent works leverage frame selection or merging strategies [55], as well as vision token merging techniques [56]. In the realm of speech processing, early methods focus on enhancing the performance of ASR [57] and speech translation [58] through speech compression techniques. More recently, FastAdaSP [40] mitigates inference overhead by performing token selection within LLMs. Concurrently, Speechprune [29] employs a token selection strategy to extend the effective speech window of Qwen2-Audio to 90

seconds for Speech Information Retrieval task. StreamUni [42] achieves real-time speech translation for long speech streams by integrating a segmentation strategy and a policy-decision module.

## 7  Conclusion

In this paper, we introduce FastLongSpeech, a novel approach that extends the capabilities of LSLMs to efficiently conduct long-speech processing. Experiments show that our method significantly reduces the computational costs and inference time in long-speech tasks, achieving better trade-offs between performance and efficiency.

## Limitations

Given the current scarcity of long-speech data, FastLongSpeech introduces an innovative dynamic compression training approach. This method leverages short-speech training data to extend the capabilities of LSLMs for long-speech processing. As long-speech training and evaluation data become more abundant in the future, FastLongSpeech will further enhance its ability to process longer speech inputs using the expanded datasets with lower costs.

## Acknowledgement

We gratefully acknowledge all the reviewers for their valuable comments and suggestions. This work was supported by the Natural Science Foundation of Beijing, China (Grant No. L257006).

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

## A  Description of LongSpeech-Eval

LongSpeech-Eval is a novel benchmark we propose for evaluating the long-speech understanding capabilities of Large Speech-Language Models (LSLMs). This benchmark presents a spoken Question-Answering (QA) task, challenging LSLMs to answer questions based on the extended speech inputs. The dataset comprises 164 samples, with an average speech duration of 132.77 seconds and a maximum duration reaching 1000 seconds.

The foundation for LongSpeech-Eval is the MultiField-En and NarrativeQA subsets from LongBench, an established long-context understanding benchmark. MultiField-En is a single-document QA dataset encompassing diverse domains, with questions and answers meticulously annotated by Ph.D. students. NarrativeQA consists of long stories along with questions posed to test reading comprehension. Our methodology for creating LongSpeech-Eval involves a rigorous multi-step process.

We first employ Llama3.1-70B-Instruct[8] to filter out samples containing numerous formulas or non-English characters, ensuring the dataset's suitability for speech synthesis and comprehension. GPT-4o [32] is utilized to summarize and polish the documents into more natural spoken forms, enhancing their suitability for speech synthesis. We then reapply Llama3.1-70B-Instruct to eliminate any samples where questions could not be adequately answered based on the spoken-form documents, ensuring the validity of the samples. Finally, we leverage the Text-to-Speech (TTS) model Orca[9] to synthesize speech from the refined spoken-form documents.

The resulting dataset combines synthesized speech with corresponding questions and answers, forming a comprehensive spoken QA benchmark.

## B  Details of Dataset

In this section, we provide a detailed description of the training and testing data.

### B.1  Training Dataset

Our training method is divided into two stages.

In the first stage, we train the CTC decoder using the CTC loss [22]. During this stage, only ASR data are used, including 960 hours of LibriSpeech [33] data and 3k hours of data sampled from MLS dataset [34].

---

[8] `https://huggingface.co/meta-llama/Llama-3.1-70B-Instruct`
[9] `https://github.com/Picovoice/orca`

In the second stage, we utilize our proposed dynamic compression training approach to train the LLM. For this stage, we use spoken QA datasets, which come from three datasets: OpenASQA [35], LibriSQA [36], and Common Voice [37]. For **OpenASQA**, we select the Open-Ended Speech AQA subset, which contains 5.9k hours of speech data. The questions and answers in this dataset are generated by GPT-3.5-Turbo and cover aspects such as spoken text, speaker gender, age, style, and emotion. For **LibriSQA**, we use the complete training set, which contains 360 hours of training data. The questions and answers in this dataset are generated by ChatGPT, with the speech data sourced from the LibriSpeech train-clean-360 subset [36]. For the **Common Voice** ASR dataset, we transform it into a spoken QA format to enhance our training set. First, we use ChatGPT to generate 200 diverse speech transcription instructions. For each ASR sample, we randomly select one instruction as the question and use the ground-truth transcription as the answer, resulting in 1.7k hours of training data.

## B.2   Evaluation Dataset

For testing, we evaluate our method on short-speech spoken QA, long-speech spoken QA, and ASR tasks. The long-speech spoken QA task corresponds to the LongSpeech-Eval benchmark, which is introduced in Appendix A.

For short-speech spoken QA, we utilize three test sets: the speech_QA_iemocap (AIR-Bench) [38], the LibriSQA test set [36], and the LibriTTS test subset from OpenASQA [35]. The speech_QA_iemocap dataset comes from the AIR-Bench benchmark and contains 200 samples. The LibriSQA test set includes 2620 samples. For the LibriTTS test subset, we select samples corresponding to the LibriTTS test-clean set from OpenASQA, keeping only the 417 samples with a speech duration longer than 15 seconds as our test set. All test sets are under 30s in duration.

For spoken dialogue understanding, we evaluate the inference efficiency of our method using speech_dialogue_QA_fisher subset [38] from AIR-Bench. This subset contains 200 samples. For this task, our method is directly applied to vanilla Qwen2-Audio, which has only undergone the first training phase of our method. This setup allows us to assess the effectiveness of our approach without requiring the training of LSLMs.

For the ASR task, we use the LibriSpeech [33] test-clean, test-other, and GigaSpeech [41] test set as our evaluation datasets. For convenience in evaluation, we convert these datasets into the spoken QA format, where the instruction for each sample is: "**Transcribe the speech to text without explanation:** ".

For emotion recognition task, We leverage the MELD dataset [39] to benchmark our method against other efficiency method [40] under diverse efficiency scenarios. FastAdaSP lowers the inference costs of Qwen2-Audio through the layer-wise dynamic reduction of speech representations within the LLM's architecture. We compare our method with FastAdaSP to demonstrate the advantage of our method in retaining information. Since we could not find the specific prompt in FastAdaSP [40], we utilize the following prompt: "**Given the Choices: [Anger, Disgust, Fear, Joy, Neutral, Sadness, Surprise]. What is the emotion in the audio?**"

## C   Experimental Details

In this section, we introduce the NTK-RoPE method in greater detail and outline the system configuration of FastLongSpeech. Our FastLongSpeech primarily leverages Qwen2-Audio. Additionally, we apply our approach to a vanilla Qwen2.5-Omni [15] model that has only undergone the first training phase. This serves to validate that our method can achieve competitive performance without altering the inherent capabilities of the model, while also demonstrating its generalizability.

NTK-RoPE extends the speech window of Qwen2-Audio to match the context length of its LLM by adjusting the Rotary Position Embedding (RoPE). **However, some samples in our LongSpeech-Eval may still exceed this extended context length. To handle these special cases, we apply our iterative fusion strategy to reduce the sequence of speech representations to fit within the prescribed context length**.

We then delineate the configuration of FastLongSpeech. The training process is in two stages. In the first stage, we utilize ASR data to train the CTC Decoder, which is a feed-forward network with

Table 6: Settings of FastLongSpeech.

| Hyperparameters | | | Settings |
|---|---|---|---|
| CTC Decoder | Model | hidden_dim
output_dim | 4096
10000 |
| | Training Details | per_device_batch_size
learning_rate
lr_scheduler | 16
2e-5
cosine |
| LSLM | Base_model | Base_model | `Qwen2-Audio-7B-Instruct` |
| | LoRA | lora_r
lora_alpha
lora_dropout
lora_target_modules | 128
256
0.05
q_proj, k_proj, v_proj, o_proj |
| | Training Details | per_device_batch_size
learning_rate
lr_scheduler | 16
2e-4
cosine |

one hidden layer. We use the SentencePiece[10] toolkit to construct the vocabulary for the training of the CTC decoder. This vocabulary is extracted from the ASR dataset. The second stage focuses on training the LLM within the LSLM using Spoken QA data. Both training stages leverage DeepSpeed[11] ZeRO-2 for optimization. Table 6 provides additional training and configuration details.

# D  Evaluation Template

In this section, we present the prompt template used for evaluating LSLMs. As shown in Figure 3, the template will be employed by the LLM to score the responses generated by the LSLMs. This scoring template is used to evaluate long-speech and short-speech spoken QA tasks.

# E  Applicability of Our Method to Vanilla LSLMs

In our FastLongSpeech framework, we extend LSLMs for long-speech processing by adopting an iterative fusion strategy and a dynamic compression training approach. As highlighted in subsection 5.2, our method not only excels in long-speech tasks but also achieves a good balance between performance and efficiency in short-speech scenarios. Therefore, this prompts us to investigate whether vanilla LSLMs can benefit from our method to effectively balance computational efficiency and generation quality, thus meeting diverse requirements across various speech processing applications. To this end, we apply the iterative fusion strategy directly to the vanilla Qwen2-Audio and vanilla Qwen2.5-Omni model.

To demonstrate the effectiveness and robustness of our method, we first extend our experiments to spoken dialogue understanding task. For this task, we conduct experiments on vanilla Qwen2-Audio using speech_dialogue_QA_fisher of AIR-Bench [38]. As shown in Table 7, our method effectively balances performance and inference efficiency. Notably, at lower compression ratios ($L$ = 200), our approach demonstrate comparable performance to the vanilla Qwen2-Audio model with a 50% reduction in computational costs. Moreover, even at a higher compression ratio of 15x ($L$=50), our method still maintains robust per-

Table 7: The experiment results on the speech_dialogue_QA_fisher subset, where "Baseline" denotes vanilla Qwen2-Audio and "Ours" denotes applying iterate fusion strategy to vanilla Qwen2-Audio.

| Method | Score ($\uparrow$) | TFLOPs ($\downarrow$) |
|---|---|---|
| Baseline | 3.95 | 11.76 |
| Ours ($L$=400) | 4.13 | 8.25 |
| Ours ($L$=200) | 3.92 | 5.46 |
| Ours ($L$=100) | 3.62 | 4.06 |
| Ours ($L$=50) | 3.16 | 3.35 |

[10]`https://github.com/google/sentencepiece`
[11]`https://github.com/deepspeedai/DeepSpeed`

I need your help to evaluate the performance of several models in the speech interaction scenario. Given a segment of speech and a related text question, the model needs to understand the speech and the question, and provide a text answer. Your task is to rate the model's responses based on the question **[Question]**, ground-truth response **[Ground-truth]** and the model's response **[Response]**. Please evaluate the faithfulness of the model's responses, and provide a score for each on a scale of 1 to 5.

**Faithfulness (1-5 points)**:
**5 points**: The model's response is completely faithful to the ground-truth response and answer the questions, covering all key information and expressed clearly and accurately.
**4 points**: The model's response is generally faithful to the ground-truth response and answer main questions, missing a few non-essential details, but the overall meaning is correct.
**3 points**: The model's response is somewhat faithful to the ground-truth response and answers the question partially, but the expression is not accurate enough or some important information is omitted, which may lead to misunderstandings.
**2 points**: The model's response differs significantly from the ground-truth response and doesn't answer the questions, with key content missing or expressed vaguely, affecting comprehension.
**1 point**: The model's response is completely unfaithful to the ground-truth response and doesn't answer the questions at all, containing errors or being entirely irrelevant, failing to convey the required information.

Below are the question, ground-truth response and model's response:
### **[Question]**: {question}
### **[Ground-truth]**: {ground_truth}
### **[Response]**: {response}

After evaluating, please output the scores in JSON format: {\"faithfulness\": faithfulness score}. You don't need to provide any explanations.

Figure 3: The prompt template for the LLM to evaluate the response of LSLMs.

formance. These findings underscore the efficacy and versatility of our iterative fusion strategy.

We further extend our approach to vanilla Qwen2.5-Omni [15], a model exhibiting superior capabilities compared to Qwen2-Audio. Specifically, we benchmark the performance of Qwen2.5-Omni against Qwen2-Audio on the speech_QA_iemocap subset of AIR-Bench. The results in Table 8 indicate that Qwen2.5-Omni, owing to its stronger speech capabilities, demonstrates superior performance across various

Table 8: The experiment results on the speech_QA_iemocap subset.

| $L$ | Qwen2-Audio | Qwen2.5-Omni |
|-----|-------------|--------------|
| 750 | 3.68 | 3.82 |
| 400 | 3.69 | 3.82 |
| 200 | 3.67 | 3.75 |

compression ratios. This demonstrates that our method achieves superior performance on more capable LSLMs, highlighting its generalizability.

## F Applicability of Our Method to Other Tasks

Additionally, we extend our experimental evaluation to the emotion recognition task and employ MELD dataset [39]. For this task, we benchmark our method against FastAdaSP [40], an approach designed for enhancing inference efficiency of Qwen2-Audio. We adopt the identical experimental setup as in the FastAdaSP, comparing performance under the same inference reduction settings. As depicted in Table 9, our method not only achieved superior performance but also reduced inference cost by 50% compared to FastAdaSP-Sparse. This underscores

Table 9: The experiments on the MELD dataset, where the results are reported in the configuration with a 50% reduction in inference cost. The performance is measured with accuracy metric.

| Method | Accuracy (%) (↑) |
|--------|------------------|
| FastAdaSP | 52.14 |
| **FastLongSpeech** | **52.95** |

Table 10: The experiments on the SPIRAL-H dataset. The performance is measured with accuracy metric.

| Method | Prune Rate (%)(↑) | Accuracy (%) (↑) |
|---|---|---|
| SpeechPrune | 60.00 | 63.77 |
| **FastLongSpeech** ($L = 750$) | **65.88** | **76.81** |

Table 11: The performance of FastLongSpeech with varying context lengths.

| **L** | 200 | 400 | 750 | 1200 | 4000 |
|---|---|---|---|---|---|
| Score (↑) | 2.47 | 2.90 | 3.55 | 3.66 | 3.59 |

the effectiveness of our approach in preserving crucial information. Furthermore, our method is complementary to the method [40] and holds potential for further improving inference efficiency through integration, a prospect we leave for future investigation.

Beyond emotion recognition, we also conduct additional experiments on the SPIRAL-H[12] dataset, a benchmark designed for long speech information retrieval. On this dataset, we follow the experimental setup [29] and compare model performance under similar speech embedding pruning rates. As shown in the table 10, our method achieves better performance with fewer speech embeddings, demonstrating its superiority and efficiency in modeling long speech inputs.

## G   Extending Maximum Context Length

We also explore the performance of FastLongSpeech on the LongSpeech-Eval dataset with varying context lengths **L**. The results are shown in Table 11. When $L$ is less than 750, the model exhibits increasing performance with longer context, as it is trained under dynamically varying compression ratios in this range. When $L$ equals 1200, although the model is not explicitly trained for this length, it still achieves strong performance, indicating good generalization beyond the training regime. When increases to 4000, performance slightly declines. This is expected, as the model is not exposed to such long contexts during training, despite having a larger speech context window. We think our FastLongSpeech can achieve better performance with longer effective context length as the long-speech training data becomes more available.

---

[12]`https://github.com/linyueqian/SPIRAL_Dataset`

