# OpenReview forum: "FastLongSpeech: Enhancing Large Speech-Language Models for Efficient Long-Speech Processing"
_NeurIPS.cc/2025/Conference — NeurIPS 2025 poster_

### Official Review · Reviewer_tmJM · 2025-06-23

**Clarity:** 3
**Significance:** 3
**Originality:** 3
**Rating:** 4
**Confidence:** 4

**Summary:**

This paper studies modelling long-input in audio/speech enabled language models (I’ll refer to audio/speech language model as ALM). The motivation is that various ALMs (e.g., Qwen2-Audio, SALMONN) aligns continuous audio representations with the textual representations of the base LLM. However, audio representations usually have much higher frame rates (e.g., 30-second audio chunk accounts for 750 tokens in Qwen2-Audio). This limits the ability of ALMs to handle long-input audios (another limitation being limited long-audio training data).

This work proposes an extractor module on top of the audio representations (before passing them to the shared representation space). The extractor module iteratively compresses the number of audio tokens, while maintaining the information. This extractor module is inspired by the CTC loss, and hence is trained in two stages where the first stage adopts the CTC loss (it identifies less informative audio frames from the CTC predictions). The proposed method is designed to extend current ALMs so that they can operate with fewer audio tokens in the shared representation space.

Experiments compare the proposed method against other audio representation compression methods, and show that this proposed method is effective. However, it should be noted that the paper doesn’t achieve extending the maximum audio context length of ALMs (e.g., beyond 30 seconds for Qwen2-Audio), but provides an effective audio representation compression could potentially allow the extension (albeit future work is required).

**Questions:**

Q1) [see weakness] Have you considered experimenting with actually extending the maximum context length? (
Q2) Would this approach be applicable to other architectures (e.g discrete token-based audio LLMs?)

**Ethical Concerns:**

["NO or VERY MINOR ethics concerns only"]

**Final Justification:**

I decided to keep my original assessment as the main title of this paper is about "long speech" -- although it has shown how this could be achieved, the paper itself hasn't actually delivered a model with a long-speech capability yet. I understand that data is one limitation; however, one could prepare/propose such long data and perhaps test it at a smaller scale. Overall, my original assessment was also leaning accept.

**Limitations:**

Yes

**Quality:**

3

**Strengths And Weaknesses:**

**Strengths**
- The idea of the iterative compression method is novel
- The audio representation compression is shown to be effective
- The paper provides LongSpeech-Eval, which could be a useful resource

**Weakness**
- The paper has not explored extending the maximum audio input context of ALMs. The authors stated that this is due to limited long audio data being available. However, I believe that obtaining such data (e.g., 60-second or say 5-minute chunks from existing raw audio data such as Yodas2) could allow the authors to validate the proposed method for actual long contexts.

Suggestion: Some work that has tried long audio input [Audio Flamingo2](https://arxiv.org/pdf/2503.03983) achieves 30 secs to 5 mins

---

> ### Author Rebuttal · Authors · 2025-07-28
>
> >Obtain such data (e.g., 60-second or say 5-minute chunks from existing raw audio data) could allow the authors to validate the proposed method for actual long contexts?
>
> Thank you for your valuable suggestion. In the future, as more long-form speech training data becomes available, our method stands to benefit from it by further enhancing the speech context window and long speech processing capabilities of speech LLMs in an efficient manner. In addition, we appreciate the reference [1] you provided and will cite it in the next version of our paper.
>
> [1] Ghosh et al. Audio Flamingo 2: An Audio-Language Model with Long-Audio Understanding and Expert Reasoning Abilities. In ICML 2025.
>
> >Experiment with actually extending the maximum context length?
>
> Following your suggestion, we explore the performance of FastLongSpeech on the LongSpeech-Eval dataset with varying context lengths $L$. The results are shown below:
>
> | LongSpeech-Eval | | | | | |
> |:--:|:--:|:--:|:--:|:--:|:--:|
> | $L$ | 200 | 400 | 750 | 1200 | 4000 |
> | **FastLongSpeech** | 2.47 | 2.90 | 3.55 | 3.66 | 3.59 |
>
> When $L \leq 750$, the model exhibits increasing performance with longer context, as it is trained under dynamically varying compression ratios in this range.\
> When $L$ equals 1200, although the model is not explicitly trained for this length, it still achieves strong performance, indicating good generalization beyond the training regime.\
> When $L$ increases to 4000, performance slightly declines. This is expected, as the model is not exposed to such long contexts during training, despite having a larger speech context window.
>
> We think our FastLongSpeech can achieve better performance with longer effective context length as the long-speech training data becomes more available.
>
> >Would this approach be applicable to other architectures, such as discrete token-based audio LLMs?
>
> We think it is feasible. In discrete token-based audio LLMs such as GLM-4-Voice [1] and Kimi-Audio [2], discrete speech units are introduced into the vocabulary to enable both understanding and generation of speech. Given that such models have demonstrated strong performance on ASR and TTS tasks, the speech embeddings in the vocabulary corresponding to speech units are also capable of encoding rich information such as **content**, prosody, emotion, and speakers. Therefore, **our content-density-guided iterative fusion strategy and dynamic compression training hold promise for application to the embedding sequences of discrete speech units as well**. Due to the limited time available during the rebuttal period, we have not been able to explore this direction yet, but we plan to investigate it further in future work based on your suggestion.
>
> [1] Zeng et al. GLM-4-Voice: Towards Intelligent and Human-Like End-to-End Spoken Chatbot. In ICLR 2025.\
> [2] KimiTeam et al. Kimi-Audio Technical Report. In arXiv:2504.18425.
>
> If our response addresses your questions and concerns, we would greatly appreciate it if you could consider raising your score.

---

> ### Comment · Reviewer_tmJM · 2025-08-04
>
> Thank you for your rebuttal! I like the paper and its contribution.
>
> However, I decided to keep my original assessment as the main title of this paper is about "long speech" -- although it has shown how this could be achieved, the paper itself hasn't actually delivered a model with a long-speech capability yet (I understand data is one limitation; however, one could prepare/propose such long data and perhaps test it at a smaller scale).

---

### Official Review · Reviewer_jHca · 2025-06-29

**Clarity:** 3
**Significance:** 3
**Originality:** 2
**Rating:** 5
**Confidence:** 4

**Summary:**

This paper proposes a frame merging strategy to compress the time dimension of the speech embeddings in a large speech LLM. The LLM is based on Qwen2 -audio model. The compression algorithm is based on the CTC decoder predictions and neighboring frame similarity. For a given audio input, a context density is computed from the non-blank CTC probability and a cosine similarity score between adjacent frames. In an iterative way, depending on the target number of frames, the most similar adjacent frames are merged into groups and the group representation is obtained by a weighted average of the embeddings in that group where the weights are based on the content density. These time-reduced embeddings then pass through the adapter and the LLM layers. Experiments investigate the effectiveness of the approach on spoken question answering (SQA) for both short and long speech signals, and ASR as well as they compare the inference cost in terms of FLOPs. The proposed method is compared against the Qwen2-audio, a random downsampling of frames, an averaging based scheme, and a ROPE based approach. On both the short-form and long-form SQA tasks, the proposed method outperforms the baselines in terms of answer quality while providing reduction in FLOPs and inference time. On the ASR task, however, the WER gets significantly higher as the compression rate increases (the target length decreases) which is expected as the task itself does not require content compression.

**Questions:**

1) In Table 5, it is not clear how the shorter audios are counted. For instance, what does L=750 mean for the utterances much shorter than 30 seconds?

2) In Table 5, for the L=100 case, it could have been better to report the WER versus test audio length (in number of embedding frames before compression) instead of an overall average because mapping a 4s audio versus a 30s-long audio with the same L would lead to a high variance in the WER per utterance.

3) The limitations section does not really mention limitations but it mentions about the potential use cases of the proposed method.

**Ethical Concerns:**

["NO or VERY MINOR ethics concerns only"]

**Final Justification:**

+++ Post-rebuttal:

It is good to see the performance breakdown based on the utterance length in the rebuttal. I am increasing my score to accept.
The observations about the compression rate make sense. I am increasing my score to accept.

**Limitations:**

Please see the question (3) above regarding the limitations section in the paper.

To my understanding, the model still chunks the audio longer than 30s into 30s chunks. It could have been beneficial to show the effectiveness of the method when we indeed provide a 30 minutes or an hour long audio as input during training.

**Quality:**

3

**Strengths And Weaknesses:**

Strengths:

Quality: Mathematical formulation and the experimental design seems to be correct. Ablation studies in Section 5.1 show the effectiveness of the design choices. Experiments show both quality improvements and reduction in inference run time. The conclusion regarding the best target length (or compression ratio) being dependent on the particular task is an expected result.

Clarity: Mostly clear.

Significance: Handling long-form speech has been an open problem even though there are several approaches to handle them successfully. Hence, this paper is relevant to the spoken LLM community as well as long-form ASR community.

Originality: Even though the technical details of the compression algorithm is not very novel (weighted average of similar consecutive), its investigation for long-form speech LLMs is novel enough.


Weaknesses:

Quality: Regarding the compression ration discussion, it might have been useful to include some other speech tasks (e.g. speech summarization) to further support the claims.

Clarity: In Table 5, it is not clear how the shorter audios are counted. For instance, what does L=750 mean for the utterances much shorter than 30 seconds? Especially, for the L=100 case, it could have been better to report the WER per test audio length (in number of embedding frames before compression) instead of an overall average.

---

> ### Author Rebuttal · Authors · 2025-07-28
>
> Thanks for your valuable suggestions and insights.
>
> >Conduct some other speech tasks (e.g. speech summarization) to further support the claims?
>
> Following your suggestion, we conduct additional experiments on the SPIRAL-H dataset, a benchmark designed for long speech information retrieval. On this dataset, we follow the experimental setup in [1] and compare model performance under similar speech embedding pruning rates. The results are presented in the table below, where Prune Rate indicates the proportion of speech embeddings removed compared to the original inputs. SpeechPrune, RAP, and RAC are baselines introduced in the referenced paper [1].
>
> | SPIRAL-H | | |
> |:--:|:--:|:--:|
> | **Methods** | **Prune Rate** ($\uparrow$) | **Accuracy** ($\uparrow$) |
> | RAP | 60 | 21.45 |
> | RAC | 60 | 35.41 |
> | SpeechPrune | 60 | 63.77 |
> | FastLongSpeech ($L$=750) | **65.88** | **76.81** |
>
> As shown in the table above, our method achieves better performance with fewer speech embeddings, demonstrating its superiority and efficiency in modeling long speech inputs. We will include these results in our paper.
>
> [1] Lin et al. SpeechPrune: Context-aware Token Pruning for Speech Information Retrieval. In ICME 2025.
>
>
> >In Table 5, it is not clear how the shorter audios are counted? It could have been better to report the WER per test audio length (in number of embedding frames before compression)?
>
> Thank you for your constructive feedback. In Table 5, when the number of original speech embeddings is shorter than the window $L$, our method does not apply compression, but instead directly feeds the original embeddings into the LLMs for understanding and inference.
>
> Following your suggestions, we conduct an additional experiment to explore the relationship between the number of pre-compression speech frames and WER, under speech window sizes of $L = 50$ and $L = 100$. The results are shown below.
>
> | LibriSpeech test-clean ($L$=50) | | || ||||
> |:--:|:--:|:--:|:--:|:--:|:--:|:--:|:--:|
> | **Number of Frames** | [1,100) | [100,200) | [200,300) | [300,400) | [400,500) | [500,600) | $\geq$ 600 |
> | **WER** ($\downarrow$) | 4.33 | 7.45 | 22.30 | 30.67 | 45.12 | 57.92 | 70.28 |
>
> | LibriSpeech test-clean ($L$=100) | | || ||||
> |:--:|:--:|:--:|:--:|:--:|:--:|:--:|:--:|
> | **Number of Frames** | [1,100) | [100,200) | [200,300) | [300,400) | [400,500) | [500,600) | $\geq$ 600 |
> | **WER**  ($\downarrow$) | 3.68 | 3.24 | 6.09 | 8.96 | 14.90 | 20.70 | 23.83 |
>
>
> These results are based on an improved system trained with a larger dataset after our paper submission.
> From the results above, we observe that as the number of original speech frames increases, the compression ratio also increases, which leads to a progressive degradation in transcription quality. However, increasing the window size ($L$) can significantly improve ASR performance, mitigating the negative effects of compression on longer inputs.
>
> >In the future, it could have been beneficial to show the effectiveness of the method when we indeed provide a 30-minute or an hour-long audio as input during training?
>
> Thanks for your suggestion. In the future, as more long-form speech training data becomes available, our method holds promise for further enhancing the speech processing capabilities of speech LLMs. By leveraging longer speech inputs during training, we aim to improve both the long-context speech and reasoning abilities of these speech LLMs, while maintaining high efficiency during inference.
>
> If our response addresses your questions and concerns, we would greatly appreciate it if you could consider raising your score.

---

> > ### Comment · Reviewer_jHca · 2025-08-04
> >
> > Thanks for responding to my questions and for sharing the WER breakdown based on the utterance length. It seems that if we downsample over 3 - 4 times, the WER significantly increases, which is a reasonable observation. I am willing to increase my score to weak accept.

---

### Official Review · Reviewer_CJSP · 2025-07-01

**Clarity:** 2
**Significance:** 2
**Originality:** 2
**Rating:** 4
**Confidence:** 4

**Summary:**

This paper proposes an extractor of speech embeddings by compressing lengthy and sparse speech representations into more compact forms using a iterative fusion based on CTC scores. With the compact representations, speech language models are enabled to handle long speech context.

**Questions:**

The effect of long context is not shown, how will different lengths of speech context affect the performance? It is expected that longer context would show better advantage of the proposed FastLongSpeech.

The distributions of lengths of speech context in LongSpeech-Eval should be provided so that the audience have better understanding of effectiveness of the proposed method.

**Ethical Concerns:**

["NO or VERY MINOR ethics concerns only"]

**Final Justification:**

Thanks for addressing my previous questions, with clear explanation and new experimental results to support the rebuttal. I have increased the score.

**Limitations:**

Yes

**Quality:**

2

**Strengths And Weaknesses:**

Strengths:
1) The proposed FastLongSpeech outperforms other compared baselines on short-speech and long-speech spoken QA tasks in terms of scores using LLM as a judge. Different lengths of compact representation consistently achieve better scores.
2) A long-speech understanding benchmark is developed using synthetic speech and used to evaluate the propose approach.

Weaknesses:
1) The evaluation on long speech context is based on synthetic speech, then how to ensure the synthetic questions and corresponding answers are reflecting correct paralinguistic characteristics? How does LongFastSpeech model the paralinguistic information in the long speech context. It is unclear how does the developed LongSpeech benchmark control the paralinguistic information during generation.
2) Sec.3.2 is the core part of the paper, but it is quite confusing to me. Eq. 5 is wrong? if T(m)=2L-1, then T(m+1)=$\lfloor{(2L-1)/2}\rfloor$ but not T(m+1)=L, according to L132-139. Also since the compression is based on the content density as defined by Eq 3, T(m+1) is not always $\lfloor T(m)/2\rfloor$ when T(m)>2L?
3) The LongSpeech-Eval benchmark is developed based on long document-based QA, however, in practice, such kind of human-LLM interaction scenarios may be limited.
4) There is a lack of comparison to a baseline that use ASR transcription of the long speech context as input, and comparison to FastAdaSP [37] and Speechprune [27] on the LongSpeech-Eval dataset.

---

> ### Author Rebuttal · Authors · 2025-07-28
>
> Thanks for your valuable feedback and suggestions.
>
> >How does the developed LongSpeech benchmark control the paralinguistic information during generation?
>
> Your question is insightful. Our LongSpeech-Eval is derived from LongBench [1], a long-text understanding benchmark that requires LLMs to answer questions based on extended context. To construct LongSpeech-Eval, we retain the original questions and answers from LongBench while filtering, rewriting, and subsequently synthesizing the context into speech.
>
> Compared to state-of-the-art TTS models such as CosyVoice 2.0 [2], Orca natively supports long-form speech synthesis and offers advantages in semantic consistency and text accuracy. In contrast, CosyVoice 2.0 often omits parts of the text when synthesizing from long input passages. Moreover, Orca also supports fine-grained control over speech, and its performance has been validated in the paper [3]. During inference, Orca leverages contextual information to identify and incorporate paralinguistic cues. Therefore, we choose Orca as our speech synthesis model.
>
> [1] Bai et al. LongBench: A Bilingual, Multitask Benchmark for Long Context Understanding. In ACL 2024.\
> [2] Du et al. CosyVoice 2: Scalable Streaming Speech Synthesis with Large Language Models. In arXiv:2412.10117.\
> [3] Fang et al. LLaMA-Omni: Seamless Speech Interaction with Large Language Models. In ICLR 2025.
>
> > How does the FastLongSpeech model the paralinguistic information in the long speech context?
>
> In the context of speech LLMs, the audio encoder extracts speech embeddings that inherently contain rich paralinguistic information, such as prosody and emotion. Our proposed method is designed to enable efficient long-context modeling of speech inputs. During the iterative fusion strategy, we perform content-guided integration of the speech embeddings based on their textual relevance, but crucially, we do not discard paralinguistic information. **This process results in compressed representations that preserve essential textual content while still retaining paralinguistic features**.
>
> As shown in Tables 8 and 9, our method achieves superior performance on speech_QA_iemocap and MELD, which are benchmarks focused on emotion recognition and emotional question answering. These results demonstrate that **although our method is guided by textual content, it effectively operates on speech embeddings, thereby maintaining a substantial amount of paralinguistic information and contributing to the model’s strong performance on affective tasks**.
>
> >Section 3.2 may be confusing to you, such as Eq. (5)?
>
> We appreciate the opportunity to provide a more detailed explanation. Section 3.2 of our paper describes the proposed iterative fusion strategy, which progressively compresses the long-form speech embeddings output by the audio encoder over multiple iterations. The goal is to reduce the sequence length step by step until it reaches the target length $L$.
>
> In each iteration, we first determine the input length $T(m)$ and output length $T(m+1)$ of the speech embeddings using Equation (5). **The value of $T(m+1)$ depends solely on $T(m)$ and is not determined by Equation (3) or the discussion in Lines 132–139**. **Once $T(m+1)$ is computed, the number of speech embeddings to be reduced in this iteration is calculated as $r(m) = T(m) - T(m+1)$**. Next, we select $r(m)$ pairs of adjacent frames for merging based on frame-wise similarity, as defined in Equation (4). The selected adjacent frames are grouped into spans, and each span is compressed into a single frame via weighted pooling using the content density in Equation (3). **After processing all spans, the number of remaining frames becomes $T(m+1)$**.
>
> Therefore, **when $T(m) = 2L - 1$, the current iteration is guaranteed to be the final one, and $T(m+1)$ must equal $L$, ensuring that the output compressed representation has the desired final length $L$**.
>
> >The human-LLM interaction scenarios in LongSpeech-Eval are potentially limited?
>
> The primary goal of LongSpeech-Eval is to assess the long-form speech understanding capability of speech LLMs, with an emphasis on broad domain coverage.
>
> To this end, we build LongSpeech-Eval upon the widely adopted LongBench [1] benchmark for long-text understanding, specifically utilizing its Single-Document QA subset. **This subset spans a diverse range of domains, including legal documents, government reports, encyclopedia entries, academic lectures, and narratives**. In practical applications, these types of content correspond to realistic scenarios such as academic presentations, public speeches, and multi-party meetings, where quick comprehension and response to long-form speech is essential.
>
> We acknowledge the importance of continuously expanding coverage and will further enhance the diversity of scenarios in future iterations of LongSpeech-Eval.
>
> [1] Bai et al. LongBench: A Bilingual, Multitask Benchmark for Long Context Understanding. In ACL 2024.
>
> >Comparison to a baseline that uses ASR transcription of the long speech context as input?
>
> Thank you for your suggestions. We implement a cascaded approach as a baseline: specifically, we first use Whisper-Large-V3 [1] to transcribe the audio, and then pass the resulting context text to Qwen-7B-Chat (since Qwen2-Audio-7B is based on Qwen-7B-Chat) for question answering.
>
> We compare this cascaded baseline with our proposed FastLongSpeech method on LongSpeech-Eval, and the results are shown in the table below:
>
> | LongSpeech-Eval | | |
> |:--:|:--:|:--:|
> | **Methods** | **Score** ($\uparrow$) | **Time (s)** ($\downarrow$) |
> |  Cascaded Method | 3.75 | 17.23 (ASR) + 1.38 (LLM) |
> |  FastLongSpeech ($L$=750) | 3.55 | 1.47 |
> | FastLongSpeech ($L$=1200) | 3.66 | 2.52 |
>
> As shown above, while our method slightly underperforms the cascaded approach in response quality, it achieves over a 7× speedup, highlighting its significant efficiency advantage for long speech processing. This quality result is not surprising, as it has also been observed that current speech LLMs tend to lag behind cascaded pipelines in certain content-based speech understanding tasks [2].
>
> We further evaluate both methods on the speech_QA_iemocap dataset, which involves emotional question answering. The results are as follows:
>
> | speech_QA_iemocap | |
> |:--:|:--:|
> | **Methods** | **Score** ($\uparrow$) |
> | Cascaded Method | 3.59 |
> | FastLongSpeech ($L$=100) | **4.15** |
>
> In this case, our method significantly outperforms the cascaded approach, highlighting the strength of an end-to-end framework in capturing emotional and paralinguistic cues from speech.
>
> [1] Radford et al. Robust Speech Recognition via Large-Scale Weak Supervision. arXiv:2212.04356.\
> [2] Wang et al. AudioBench: A Universal Benchmark for Audio Large Language Models. In NAACL 2025.
>
> >Comparison to FastAdaSP [37] and Speechprune [27] on the LongSpeech-Eval?
>
> The FastAdaSP method does not extend the long-speech capabilities of speech LLMs; instead, it focuses solely on improving inference efficiency. Therefore, we did not include it as a baseline on LongSpeech-Eval.
>
> As for SpeechPrune, we are unable to conduct experiments on LongSpeech-Eval since the method is not open-sourced. **Therefore, to provide a comparison, we conduct experiments on the SPIRAL-H benchmark [1], which is introduced in the paper of SpeechPrune for long speech information retrieval**. The results are presented in the table below, where Pruning Rate indicates the proportion of speech embeddings removed compared to the original inputs.
>
> | SPIRAL-H | | |
> |:--:|:--:|:--:|
> | **Methods** | **Prune Rate** ($\uparrow$) | **Accuracy** ($\uparrow$) |
> | SpeechPrune |  60 | 63.77 |
> | FastLongSpeech ($L$=750) | **65.88** | **76.81** |
>
> As shown in the table above, our method achieves better performance with fewer speech embeddings, demonstrating its superiority and efficiency in modeling long speech inputs. We will include these results in our paper.
>
> [1] Lin et al. SpeechPrune: Context-aware Token Pruning for Speech Information Retrieval. In ICME 2025.
>
> >How will different lengths of speech context affect the performance?
>
> To investigate how our method performs under different context lengths, we conduct experiments on LongSpeech-Eval with varying values of $L$. The results are shown below:
>
> | LongSpeech-Eval | | | | | |
> |:--:|:--:|:--:|:--:|:--:|:--:|
> | $L$ | 200 | 400 | 750 | 1200 | 4000 |
> | **FastLongSpeech** | 2.47 | 2.90 | 3.55 | 3.66 | 3.59 |
>
> When $L \leq 750$, the model exhibits increasing performance with longer context, as it is trained under dynamically varying compression ratios in this range.\
> When $L$ equals 1200, although the model is not explicitly trained for this length, it still achieves strong performance, indicating good generalization beyond the training regime.\
> When $L$ increases to 4000, performance slightly declines. This is expected, as the model is not exposed to such long contexts during training, despite having a larger speech context window.
>
> >The distributions of lengths of speech context in LongSpeech-Eval should be provided?
>
> Thank you for your suggestion. The distribution of speech durations, measured in seconds, is shown in the table below:
>
> | LongSpeech-Eval | | | | | | |||
> |:--:|:--:|:--:|:--:|:--:|:--:|:--:|:--:|:--:|
> | **Duration Range** | [50, 70) | [70, 80) | [80, 90) | [90, 100) | [100, 110)  | [110, 130) | [130, 170) | $\geq$170 |
> | **Percent** | 9.15 | 21.39 | 17.68 | 12.80 | 9.76 | 9.76 | 8.54 | 7.93 |
>
> **If our response addresses your questions and concerns, we would greatly appreciate it if you could consider raising your score.**

---

### Official Review · Reviewer_BctC · 2025-07-03

**Clarity:** 3
**Significance:** 3
**Originality:** 3
**Rating:** 5
**Confidence:** 3

**Summary:**

This paper introduces FastLongSpeech, a framework designed to extend Large Speech-Language Models (LSLMs) for efficient processing of long-form speech without requiring dedicated long-speech training data. The approach builds on Qwen2-Audio and incorporates two main technical contributions: (1) an iterative fusion strategy that compresses long speech sequences by merging similar adjacent frames based on content density measured via CTC output, and (2) a dynamic compression training method that adapts models to varying compression ratios using only short-speech data. The authors also contribute LongSpeech-Eval, a benchmark for evaluating long-speech understanding capabilities. Experimental results demonstrate that FastLongSpeech achieves competitive performance on both short and long speech tasks while significantly reducing computational costs compared to baseline approaches.

**Questions:**

Comparison with Established Methods: Why weren't established long-sequence modeling techniques like chunking with overlap, hierarchical attention, or other compression methods compared as baselines? A comparison with these approaches would strengthen the paper's claims about the effectiveness of the proposed method.

**Ethical Concerns:**

["NO or VERY MINOR ethics concerns only"]

**Final Justification:**

Thank you for the response. I've read the authors' responses and appreciate their reflection on the paper. I will keep my score and maintain my recommendation.

**Limitations:**

The authors provide a brief limitations section but could expand on several critical points.

**Paper Formatting Concerns:**

The paper generally follows NeurIPS formatting guidelines well.

**Quality:**

3

**Strengths And Weaknesses:**

**Strengths:**

- **Practical Significance**: The work addresses a genuine limitation in current LSLMs - the inability to efficiently process long-form speech. This is practically important for applications like lecture transcription, long-form audio analysis, and extended conversations.

- **Technical Innovation**: The iterative fusion strategy is well-motivated, using content density from CTC outputs to guide compression decisions. The dynamic compression training approach cleverly leverages short-speech data to transfer capabilities to long-speech scenarios without requiring expensive long-speech datasets.

- **Comprehensive Evaluation**: The experimental setup is thorough, covering multiple tasks (spoken QA, ASR, emotion recognition) and providing both performance and efficiency metrics. The creation of LongSpeech-Eval benchmark is a valuable contribution.

- **Clear Methodology**: The paper clearly explains the iterative fusion algorithm and training procedure, making the approach reproducible.

**Weaknesses:**

- **Limited Baseline Comparisons**: The paper lacks comparison with other established long-sequence modeling techniques beyond NTK-RoPE. Methods like sliding window attention or other compression strategies from the broader literature could provide stronger baselines.

- **Synthetic Benchmark Limitations**: LongSpeech-Eval relies on synthesized speech from text documents, which may not capture the complexities and variabilities of natural long-form speech. The benchmark's validity for real-world scenarios is questionable.

- **Content Density Assumption**: The approach assumes CTC-based content density is a reliable indicator of information importance, but this may not hold across all speech types (e.g., prosodic information, speaker characteristics, emotional content may be important but have low text density).

- **Limited Scale Evaluation**: Experiments are conducted only on 7B parameter models. The scalability and effectiveness of the approach on larger, more capable models remains unclear.

---

> ### Author Rebuttal · Authors · 2025-07-28
>
> Thank you very much for your insightful comments and the time you spent on reviewing.
>
> >Consider comparing with methods like chunking with overlap, sliding window attention, to provide more comparisons?
>
> Thank you for your valuable comments. There are two main reasons why our comparisons primarily focus on compression-based and expansion-based methods for extending speech LLM input windows, rather than the mentioned established text-based methods.
>
> First, the complexity of speech inputs poses unique challenges. Unlike text, the speech inputs are inherently more sparse in information density, and sentence boundaries are often ambiguous and difficult to determine. **When directly applying traditional long-sequence modeling methods involved chunking, the segmentation process can lead to semantic discontinuity and misinterpretation**.
>
> Second, the traditional long-context methods are designed on top of already strong language models, aiming to further extend their ability to handle ultra-long textual sequences. In contrast, the long-speech capabilities of current speech LLMs are still limited, which capture much less information per unit input than their text counterparts. Therefore, **our work instead focuses on enhancing the inherent speech capability of speech LLMs themselves**. Furthermore, **our approach is complementary to your proposed techniques and can potentially be combined with them in future work to further improve long speech modeling capabilities**.
>
> Following your suggestions, **we also implement two additional variants to provide a broader set of baselines**:
>
> **Word-Bound**, which uses CTC to detect word boundaries and then aggregates representations belonging to the same word.\
> **Blank-First**, which prioritizes merging segments with blank (non-linguistic) predictions.
>
> As shown in the table below, our method achieves superior performance, demonstrating its effectiveness in improving long-context modeling for speech.
> | LongSpeech-Eval | | | |
> |:--:|:--:|:--:|:--:|
> | **Method** | FastLongSpeech | Word-Bound | Blank-First |
> | **Score** ($\uparrow$) | 3.55 | 2.21 | 3.067 |
>
> >Synthetic Benchmark may not capture the complexities and variabilities of natural long-form speech?
>
> Thank you for your insightful question. To ensure the quality of the speech in our proposed LongSpeech-Eval benchmark, we take several measures to guarantee the quality of the benchmark.
>
> On the one hand, we apply a filtering process using LLMs to remove samples that are unsuitable for speech synthesis. For the remaining samples, we leverage GPT-4o, an advanced language model, to rewrite and improve the textual content. Finally, we conduct manual verification to further ensure the quality and fluency of the text.
>
> On the other hand, we employ a high-quality commercial TTS system, Orca, to perform speech synthesis. Orca supports long-form speech generation and allows fine-grained control over the speech. Therefore, we can obtain good complexities and variabilities of long-form speech.
>
> >Content density assumption may not hold across all speech types (e.g., prosodic information, speaker characteristics, emotional content)?
>
> Thank you for your insightful comments. Compared to paralinguistic features such as prosody, textual content is inherently more information-dense and quantifiable. This characteristic motivates our design choice to develop a speech compression strategy guided by textual content density.
>
> **It is worth emphasizing that while our approach leverages content-aware compression, it fundamentally differs from traditional cascaded pipelines by preserving significant paralinguistic information**. The experimental results of FastLongSpeech on speech emotion understanding tasks (speech_QA_iemocap and MELD) demonstrate superior performance compared to alternatives, confirming that our method effectively retains and utilizes crucial paralinguistic cues.
>
> Looking forward, as speech processing research advances, our framework can be readily extended to incorporate more diverse and advanced speech information indicators. This flexibility creates opportunities for further improvements in capturing both linguistic content and paralinguistic elements in long-form speech processing.
>
> >Experiments are conducted only on 7B parameter models?
>
> Thanks for your suggestions. Given the real-time and efficiency requirements of speech processing, current open-source speech LLMs are typically around the scale of 7B parameters. However, our proposed method is not limited to a specific model. While we primarily conduct experiments on Qwen2-Audio, we have also demonstrated its effectiveness on Qwen2.5-Omni in Table 8, which supports the generalizability of our approach.
>
> As computing resources continue to advance, we expect larger-scale speech LLMs to become available. We plan to extend and validate our method on these larger models in future work.

---

> > ### Comment · Reviewer_BctC · 2025-08-08
> > **Comments to the Authors**
> >
> > Thanks for the authors' rebuttal. I think the authors' responses clarified most of my comments on the weaknesses, and I would like to keep my score as accept.

---

### Decision · Program_Chairs · 2025-09-17

**Decision:**

Accept (poster)

**Comment:**

This paper introduces FastLongSpeech, a framework that extends LSLMs to long-form speech without dedicated training data. It combines CTC-based iterative fusion for sequence compression with dynamic compression training using short-speech data, and contributes LongSpeech-Eval for evaluation. Experiments show competitive performance with lower computational cost.

In rebuttal, the authors addressed reviewers’ concerns by adding results for Word-Bound/Blank-First, reporting on additional datasets (SPIRAL-H), and analyzing performance by sequence length. All reviewers found the method novel, practical, and publication-worthy.

Due to all of the above I recommend acceptance. I strongly encourage the authors to incorporate the reviewers’ suggestions—particularly the additional results and clarifications—into the final version, doing so will significantly enhance the quality and impact of the paper.